# Characterization of the Elastoplastic Response of Low Zn-Cu-Ti Alloy Sheets Using the CPB-06 Criterion

**DOI:** 10.3390/ma12193072

**Published:** 2019-09-20

**Authors:** Francisco Alister, Diego Celentano, Javier Signorelli, Pierre-Olivier Bouchard, Daniel Pino, Marcela Cruchaga

**Affiliations:** 1Department of Mechanical and Metallurgical Engineering, Pontificia Universidad Católica de Chile, Avenida Vicuña Mackenna 4860, Macul 7820436, Chile; 2Instituto de Física de Rosario (UNR-CONICET), Ocampo y Esmeralda, Rosario S2000EZP, Santa Fe, Argentina; signorelli@ifir-conicet.gov.ar; 3MINES ParisTech, PSL Research University, CEMEF-Centre de mise en forme des matériaux, CNRS UMR 7635, CS 10207 rue Claude Daunesse, 06904 Sophia Antipolis Cedex, France; pierre-olivier.bouchard@mines-paristech.fr (P.-O.B.); daniel.pino_munoz@mines-paristech.fr (D.P.); 4Department of Mechanical Engineering, Universidad de Santiago de Chile, Avenida Bernardo O’Higgins 3363, Estación Central 9170020, Chile; marcela.cruchaga@usach.cl

**Keywords:** plastic anisotropy, CPB-06 yield criterion, Zinc alloys

## Abstract

Unlike other HCP metals such as titanium and magnesium, the behavior of zinc alloys has only been modeled in the literature. For the low Zn-Cu-Ti alloy sheet studied in this work, the anisotropy is clearly seen on the stress-strain curves and Lankford coefficients. These features impose a rigorous characterization and an adequate selection of the constitutive model to obtain an accurate representation of the material behavior in metal forming simulations. To describe the elastoplastic behavior of the alloy, this paper focuses on the material characterization through the application of the advanced Cazacu-Plunket-Barlat 2006 (CPB-06 for short) yield function combined with the well-known Hollomon hardening law. To this end, a two-stage methodology is proposed. Firstly, the material characterization is performed via tensile test measurements on sheet samples cut along the rolling, diagonal and transverse directions in order to fit the parameters involved in the associate CPB-06/Hollomon constitutive model. Secondly, these material parameters are assessed and validated in the simulation of the bulge test using different dies. The results obtained with the CPB-06/Hollomon model show a good agreement with the experimental data reported in the literature. Therefore, it is concluded that this model represents a consistent approach to estimate the behavior of Zn-Cu-Ti sheets under different forming conditions.

## 1. Introduction

Zinc is commonly used as a corrosion-resistant coating. However, it is also produced as thin sheets, mainly used in architecture and construction as roofing material, rain gutters and decorative products. In addition to its corrosion resistance property, zinc shows high malleability, ductility and a high quality and durable surface finish. Despite these wide uses, there is a lack of studies and information with respect to zinc sheet formability, in which the high *c/a* ratio may lead to a marked and evolving anisotropy in the plane of the blank as a consequence of the texture modification [1,2,3,4].

Zinc has a Hexagonal Close Packed structure (HCP), for which the rolling process leads to a strong texture, and slight local changes in the material induced by the manufacturing process (non-homogeneous cooling rates, local microsegregation of alloys, etc.) often generate significant modifications of the microstructure. This material complexity leads to a high variability of the strain and stress responses even on different samples over the same direction as shown in [5,6]. For HCP metals, the rolling process produces an alignment of the *c*-axis normal to the rolling plane with a deviation of approximately 25°, inducing a high anisotropy in the sheet [7]. Specifically for the Zn-Cu-Ti alloy, the relation between the texture from the rolling process and the bendability at different temperatures is discussed in [8]. Additionally, the evolution of texture during a rolling process with an 80% reduction in thickness is reported in [9] for the Zn-Cu-Ti alloy, comparing the texture components to those predicted by the Taylor evolution model.

Moreover, HCP materials show a Strength Differential (SD) effect due to the presence of the twining deformation mechanism. This process is asymmetric and exhibits different behavior in tension and compression [10], so the yield cannot be predicted with symmetric functions for all of the expected forming conditions. Further studies were carried out on zinc alloys to define its formability via polycrystal models and necking criteria, such as that known as the Marciniack-Kuczynski approach [6,11,12].

To accurately describe the material behavior under general forming conditions, diverse anisotropic yield functions, thoroughly described in [13], have been developed in recent decades. Until now, the formability of zinc sheets has been driven by the use of constitutive models based mainly on the Hill-48 yield criterion combined with the Hollomon or Swift hardening laws, where both the yield function and hardening law are loading angle dependent [1,5,7,14,15,16]. The works cited made use of a “Fiber Vector”, defined according to the direction of the major strain [17]. This allows the calculation of the yield function and hardening law coefficients for any direction in general, and in particular for the rolling (RD), diagonal (DD) and transverse (TD) directions. Although this approach has been demonstrated to have a good agreement between experimental and numerical results, it requires a specific expression for each tested direction, and in addition, it is unable to characterize the SD effect, which can be seen on several HCP metals. On the other hand, new asymmetrical yield functions have been developed in order to completely define the HCP behavior using a phenomenological approach [18]. One of these functions is the Cazacu-Plunket-Barlat 2006—i.e., the CPB-06—criterion [19,20], which is based on both the generalization of the Barlat-96 function [21] and the linear transformation of the Cauchy stress tensor proposed in [22]. The CPB-06 criterion introduces an asymmetry coefficient to account for the SD effect. Although this yield function was specifically formulated for HCP metals, it is flexible enough to model FCC and BCC materials. The CPB-06 criterion was firstly published in [19] and described later in detail in [20]. Many applications of this yield function can be found for titanium, magnesium and zirconium alloys [20,23,24,25,26,27,28,29], but none for zinc alloys.

In this work, the associated form of the CPB-06 yield criterion together with the Hollomon hardening law are implemented in a finite element code for the estimation of the elastoplastic response of the low Zn-Cu-Ti alloy. To this end, a two-stage methodology is proposed. Firstly, the material characterization is performed via tensile test measurements available in [1]. As already mentioned, the material response associated with RD and TD samples show notorious differences in hardening, increasing its value from RD to TD with intermediate values for DD samples. Moreover, the fracture strain is drastically reduced from RD to TD where, in addition, the Lankford coefficients are all less than one and significantly different for the three directions. The hardening coefficients were calibrated for RD, while the CPB-06 coefficients were fitted with the use of an error minimization function that considers not only the stress-strain curves along DD and TD, but also the Lankford coefficients in RD, DD and TD. Secondly, these material parameters are assessed in the simulation of the bulge test using different dies to validate the numerical model.

## 2. Materials and Methods

### 2.1. Material

The material used in this work is the low Zn-Cu-Ti alloy commercially known as Zn-20. The RD, DD and TD tensile samples were gathered from cold-rolled sheets of 1.0 mm thickness tested at a strain rate of 0.007 s^−1^ [1]. The experimental true stress-strain tensile curves obtained and reported in [1] are presented in Figure 1.

The mechanical properties, i.e., yield strengths and Lankford coefficients, of the Zn-20 alloy sheet are presented in Table 1. The Young modulus and Poisson ratio with respective values of 127.7 GPa and 0.23 were taken from [30]. These data are used in the fitting procedure to be presented in Section 2.3.

### 2.2. CPB-06/Hollomon Elastoplastic Model

The constitutive model used in this work is defined in the context of the associated flow rule and rate-independent plasticity with the standard elastoplastic strain decomposition [31]. It was assumed that RD and TD are aligned with the *x* and *y* axes in the material reference system; thus, the *z* axis defines the out-of-plane component. The CPB-06 yield criterion adopted to describe the material response is written as [19,20]:(1)$$F_{\left(\sigma ,{\bar{\varepsilon }}_{p}\right)}={\bar{\sigma }}_{\left(\sigma \right)}-Y_{\left({\bar{\varepsilon }}_{p}\right)}=0$$ where $\bar{\sigma }$ is the equivalent stress, σ is the Cauchy stress tensor, Y is the isotropic hardening stress and ${\bar{\varepsilon }}_{p}$ is the equivalent plastic strain. The equivalent stress is given by:(2)$${\bar{\sigma }}_{\left(\sigma \right)}={\left(\frac{f_{\left(\Sigma \right)}}{f_{\left(\gamma \right)}}\right)}^{\frac{1}{a}}$$
such that $f_{\left(\chi \right)}$, for χ=Σ or χ= γ, is defined as:(3)$$f={\left(\left|\chi_{1}\right|-k\chi_{1}\right)}^{a}+{\left(\left|\chi_{2}\right|-k\chi_{2}\right)}^{a}+{\left(\left|\chi_{3}\right|-k\chi_{3}\right)}^{a}$$ where *a* is the degree of homogeneity, $\Sigma_{i}$ are the principal components of the transformed stress tensor, $\gamma_{i}$ are the modified anisotropic coefficients and *k* is the asymmetry parameter (related, as already mentioned, to the SD effect). The reported expression for the transformed stress tensor Σ is given by:(4)$$\left(\begin{array}{c} \Sigma_{xx}\\ \Sigma_{yy}\\ \Sigma_{xy}\\ \Sigma_{zz}\\ \Sigma_{xz}\\ \Sigma_{yz} \end{array}\right)=\left(\begin{array}{c} L_{11}\\ L_{12}\\ 0\\ L_{14}\\ 0\\ 0 \end{array}\begin{array}{c} L_{12}\\ L_{22}\\ 0\\ L_{24}\\ 0\\ 0 \end{array}\begin{array}{c} 0\\ 0\\ L_{33}\\ 0\\ 0\\ 0 \end{array}\begin{array}{c} L_{14}\\ L_{24}\\ 0\\ L_{44}\\ 0\\ 0 \end{array}\begin{array}{c} 0\\ 0\\ 0\\ 0\\ L_{55}\\ 0 \end{array}\begin{array}{c} 0\\ 0\\ 0\\ 0\\ 0\\ L_{66} \end{array}\right){\left(\begin{array}{c} \sigma_{xx}^{\prime }\\ \sigma_{yy}^{\prime }\\ \sigma_{xy}^{\prime }\\ \sigma_{zz}^{\prime }\\ \sigma_{xz}^{\prime }\\ \sigma_{yz}^{\prime } \end{array}\right)}_{m}$$ where the components of tensor L are the anisotropic coefficients and $\sigma_{m}^{\prime }$ is the deviatoric part of the Cauchy stress tensor expressed in the material reference system. The modified anisotropic coefficients $\gamma_{i}$ are:(5)$$\gamma_{1}=\left(\frac{2}{3}L_{11}-\frac{1}{3}L_{12}-\frac{1}{3}L_{14}\right)$$
(6)$$\gamma_{2}=\left(\frac{2}{3}L_{12}-\frac{1}{3}L_{22}-\frac{1}{3}L_{24}\right)$$
(7)$$\gamma_{3}=\left(\frac{2}{3}L_{14}-\frac{1}{3}L_{24}-\frac{1}{3}L_{44}\right)$$

Moreover, the hardening behavior is described by means of the Hollomon power law written for RD as:(8)$$Y_{\left({\bar{\varepsilon }}_{p}\right)}=K{\left({\bar{\varepsilon }}_{0}+{\bar{\varepsilon }}_{p}\right)}^{n}$$ where *K* is the strength coefficient, *n* is the hardening exponent (note that in this context, unlike other approaches [1,7,14,16,17], these two coefficients are only defined for RD), and with ${\bar{\varepsilon }}_{0}={\left(\frac{\sigma_{yp}{}^{RD}}{K}\right)}^{\frac{1}{n}}$, $\sigma_{yp}{}^{RD}$ being the yield strength for RD (see Table 1). In addition, the rate of the equivalent plastic strain is $\dot{{\bar{\varepsilon }}_{p}}=\frac{\sigma_{m}:{\dot{\varepsilon }}_{p}}{\bar{\sigma }}$, such that $\varepsilon_{p}$ is the plastic strain tensor whose rate obeys the classical (objective, i.e., frame-indifferent) associated flow rule ${\dot{\varepsilon }}_{p}=\dot{\lambda }\frac{\partial \bar{\sigma }}{\partial \sigma_{m}}$, where $\dot{\lambda }$ is the plastic consistency parameter.

This model was implemented in an in-house finite element code with a radial-return scheme based on the Newton-Raphson iterative method [31]. The proposed model is used to describe different strain path-dependent behaviors in a complete set of the bulge test. The computed numerical results show good agreement with the experiments, as will be discussed in Section 4. Additionally, the present work improves previous referenced studies on Zn-Cu-Ti sheet formability, by allowing the fitting of all directions with the use of a unique set of parameters for both the yield function and Lankford coefficients.

### 2.3. Fitting Procedure via the Tensile Test 

The fitting procedure is based on the analytical expression for the stress and strain behavior on the unidirectional tensile test adopting the plane stress assumption. Thus, only $\sigma_{xx}^{}$, $\sigma_{xy}^{}$ and $\sigma_{yy}^{}$ are different from zero. The steps involved in the methodology, summarized by the flow diagram in Figure 2, are described below.

#### 2.3.1. Data Preparation

The experimental data was considered until the Ultimate Tensile Stress (UTS) in the axial true stress-true strain ($\sigma_{\theta {}^{\circ}}^{exp}-\varepsilon_{\theta {}^{\circ}}^{exp}$) curves of the θ samples (i.e., θ = 0°, 45° and 90° for RD, DD and TD, respectively), for which a homogeneous state is assumed [32]. For simplicity, the same number of experimental ($\sigma_{\theta {}^{\circ}}^{exp}-\varepsilon_{\theta {}^{\circ}}^{exp}$) values *m* were considered for the curves of the three samples. To obtain the plastic component of the axial strain $\varepsilon_{p,\theta {}^{\circ}}^{exp}$ for a stress beyond the yield strength, a simple decomposition was used:(9)$$\varepsilon_{p,\theta {}^{\circ}}^{exp}=\varepsilon_{\theta {}^{\circ}}^{exp}-\frac{\sigma_{\theta {}^{\circ}}^{exp}}{E}$$ where E is the Young’s modulus.

#### 2.3.2. Hardening Fitting

The hardening parameters (*K* and *n*) were obtained through the minimization of the following objective function:(10)$$Error_{Y}=\overset{m}{\underset{i=1}{\sum }}{\left(\frac{\sigma_{0{}^{\circ}}^{num}}{\sigma_{0{}^{\circ}}^{exp}}-1\right)}_{i}^{2}$$ where $\sigma_{0{}^{\circ}}^{num}$ is the numerical axial stress for the RD sample computed with the expression of $\sigma_{\theta {}^{\circ}}^{num}$ given in Equation (16).

#### 2.3.3. CPB-06 Fitting

The objective function proposed in [33] is also used here to obtain, through its minimization, the parameters involved in the CPB-06 model. A symmetric material response, i.e., *k* = 0, is assumed, since there is no experimental evidence of twining for this alloy for low strain rates. Moreover, *L_11_* = 1 was chosen [20,23,24,25,26,27,28,29], while *L_55_* and *L_66_* were also set to 1 due to the unavailability in this study of experimental results associated with the out-of-plane stress components. In summary, the CPB-06 parameters to be obtained are six *L* coefficients and exponent *a*. The objective function is written as:(11)$$Error_{L}=W_{45{}^{\circ}}^{T}\overset{m}{\underset{i=1}{\sum }}{\left(\frac{\sigma_{45{}^{\circ}}^{num}}{\sigma_{45{}^{\circ}}^{exp}}-1\right)}_{i}^{2}+W_{90{}^{\circ}}^{T}\overset{m}{\underset{i=1}{\sum }}{\left(\frac{\sigma_{90{}^{\circ}}^{num}}{\sigma_{90{}^{\circ}}^{exp}}-1\right)}_{i}^{2}+W_{0{}^{\circ}}^{R}\overset{m}{\underset{i=1}{\sum }}{\left(\frac{R_{0{}^{\circ}}^{num}}{R_{0{}^{\circ}}^{exp}}-1\right)}_{i}^{2}\;+W_{45{}^{\circ}}^{R}\overset{m}{\underset{i=1}{\sum }}{\left(\frac{R_{45{}^{\circ}}^{num}}{R_{45{}^{\circ}}^{exp}}-1\right)}_{i}^{2}+W_{90{}^{\circ}}^{R}\overset{m}{\underset{i=1}{\sum }}{\left(\frac{R_{90{}^{\circ}}^{num}}{R_{90{}^{\circ}}^{exp}}-1\right)}_{i}^{2}$$ where $R_{\theta {}^{\circ}}^{exp}$ and $R_{\theta {}^{\circ}}^{num}$ are the experimental and numerical Lankford coefficients of a θ sample, respectively, and *W* is a weighting factor. For simplicity, the weights *W* were set to 1 in this work for the five terms of Equation (11).

The fitting routine, to minimize Expressions (10) and (11), is based on the non-linear Levenberg-Marquardt algorithm.

*Numerical Stress*$\sigma_{\theta {}^{\circ}}^{num}$:

The numerical stress term is obtained from the general form of the equivalent stress given by Equation (2), where $f_{\left(\Sigma \right)}$ can be written for a uniaxial tensile loading in the form of:(12)$$f_{\left(\Sigma \right)}=\sigma_{\theta {}^{\circ}}^{exp}{}^{a}f_{\left(\varphi \right)}=\sigma_{\theta {}^{\circ}}^{exp}{}^{a}\;\left[{\left(\left|\varphi_{1}\right|-k\varphi_{1}\right)}^{a}+{\left(\left|\varphi_{2}\right|-k\varphi_{2}\right)}^{a}+{\left(\left|\varphi_{3}\right|-k\varphi_{3}\right)}^{a}\right]$$ where the expressions for $\varphi_{1},{\;\varphi }_{2}$ and${\;\varphi }_{3}$ are:(13)$$\varphi_{1}=\left(\frac{2}{3}L_{11}-\frac{1}{3}L_{12}-\frac{1}{3}L_{14}\right)\cos^{2}\theta {}^{\circ}+\left(-\frac{1}{3}L_{11}+\frac{2}{3}L_{12}-\frac{1}{3}L_{14}\right)\sin^{2}\theta {}^{\circ}$$
(14)$$\varphi_{2}=\left(\frac{2}{3}L_{12}-\frac{1}{3}L_{22}-\frac{1}{3}L_{24}\right)\cos^{2}\theta {}^{\circ}+\left(-\frac{1}{3}L_{12}+\frac{2}{3}L_{22}-\frac{1}{3}L_{24}\right)\sin^{2}\theta {}^{\circ}$$
(15)$$\varphi_{3}=\left(\frac{2}{3}L_{14}-\frac{1}{3}L_{24}-\frac{1}{3}L_{44}\right)\cos^{2}\theta {}^{\circ}+\left(-\frac{1}{3}L_{14}+\frac{2}{3}L_{24}-\frac{1}{3}L_{44}\right)\sin^{2}\theta {}^{\circ}$$

From (12), it can be seen that for θ°=0 (RD), $f_{\left(\varphi \right)}$ becomes $f_{\left(\gamma \right)}$ and ${\bar{\sigma }}_{\left(\sigma \right)}=\sigma_{0{}^{\circ}}$ is fulfilled. In addition, $\sigma_{\theta {}^{\circ}}^{num}$ can be written as:(16)$$\sigma_{\theta {}^{\circ}}^{num}=Y_{\left({\bar{\varepsilon }}_{p}\right)}{\left(\;\frac{f_{\left(\gamma \right)}}{f_{\left(\varphi \right)}}\right)}^{\frac{1}{a}}$$ where $\frac{f_{\left(\gamma \right)}}{f_{\left(\varphi \right)}}$ includes the set of *L* coefficients to be fitted.

*Numerical Lankford Coefficients*$R_{\theta {}^{\circ}}^{num}$:

Considering the inherent plastic incompressibility of the model, the numerical Lankford coefficients are written as:(17)$$R_{\theta {}^{\circ}}^{num}=-\frac{\frac{\partial \;\bar{\sigma }}{\partial \sigma_{yy}^{r}}}{\frac{\partial \;\bar{\sigma }}{\partial \sigma_{xx}^{r}}+\frac{\partial \;\bar{\sigma }}{\partial \sigma_{yy}^{r}}}$$ where the superscript *r* denotes the tensile test reference system such that the sample is loaded in the *x* direction.

Although the uniaxial test is very important, it is also relevant to assess the proposed model under loading conditions that are more representative of real applications. For this reason, the bulge test will be used to study the mechanical response of the material under biaxial loading.

### 2.4. Numerical Simulations of the Bulge Test

According to the bulge tests carried out in [1], three different dies with the following minor to major axis ratios β were used: 1.00 (equibiaxial), 0.50 and 0.33. For the β = 0.5 and β = 0.33 dies, samples with the major axis aligned with RD and TD were considered (the largest of the three dies was 120 mm). Therefore, five simulations were performed in order to replicate the experimental strain paths reported in [1].

The complete domain was meshed with three sub-sets: the sheet sample, the die and the sheet contact interface. The die was assumed to be rigid. For the sheet, 10800 trilinear 8-noded hexahedral elements with B-bar integration to avoid numerical locking [31] were used (considering 6 elements along the thickness), while the die and contact interface were discretized with bilinear 4-noded quadrilateral elements, 2160 for the die and 3600 for the interface. The geometrical models and finite element meshes of the bulge test for the different analyzed dies are plotted in Figure 3.

As in the experiment, an internal pressure was prescribed on one side of the sheet, with displacements restrained at the edges of the sheet. Coulomb friction is considered with a friction coefficient value of 0.3 between the sheet and the die for all simulations.

## 3. Results

### 3.1. Fitting Procedure

The obtained Hollomon and CPB-06 fitted coefficients are respectively presented in Table 2 and Table 3.

The adjusted true stress-strain curves, based on the CPB-06/Hollomon model, are displayed in Figure 4.

The numerical Lankford coefficients obtained with the parameters reported in Table 2 and Table 3 are summarized in Table 4.

The error of the fitting procedure in the true stress-strain curves and Lankford coefficients for the three test directions can be assessed through the Root Mean Square Error (RMSE) given by the expressions:(18)$$E_{\sigma }=\sqrt{\frac{1}{m}\overset{m}{\underset{i=1}{\sum }}{\left(\sigma_{\theta {}^{\circ}}^{num}-\sigma_{\theta {}^{\circ}}^{exp}\right)}^{2}{}_{i}}$$
(19)$$E_{R}=\sqrt{\frac{1}{m}\overset{m}{\underset{i=1}{\sum }}{\left(R_{\theta {}^{\circ}}^{num}-R_{\theta {}^{\circ}}^{exp}\right)}^{2}{}_{i}}$$

The obtained RMSEs for the true stress-strain curves and Lankford coefficients for the three test directions are shown in Table 5.

Figure 5 shows the plane stress yield envelope in the $\sigma_{xx}$ and $\sigma_{yy}$ plane (with $\sigma_{xy}=0$) at the initiation of yielding for the von Mises, Hill-48 and CPB-06 criteria. The Hill-48 function is computed based on the *R* values shown in Table 1.

### 3.2. Bulge Test

The experimental and numerical strain paths on the major and minor strains diagram for the different dies and sample orientations are plotted in Figure 6 (the results from the tensile tests are also included for completeness). The experimental measurements correspond to those reported in [1]. The numerical results were gathered from the central element of the external side of the sheet.

## 4. Discussion

The complexity of the anisotropy shown by zinc alloys requires the use of more elaborate elastoplastic constitutive laws. Thanks to the recent advances of the material science community, we can dispose right now of a rather large amount of the various constitutive laws that can be used to study anisotropic materials like zinc alloys. These tools vary in complexity, and of course in precision. As one might expect, more complex constitutive relationships often convey a better precision, but also, in a general way, it can be said that the more complex the material, the larger the number of material parameters that should be identified [34,35,36]. Needless to say, that large number of material parameters relies on complex and expensive experimental campaigns, which often do not meet the requirements of competitive industries. Additionally, the identification process of these material parameters is carried out through the use of inverse analysis tools, typically leading to ill-posed problems. The challenge consists in obtaining a balance between complexity and precision.

As already mentioned, previous studies on zinc alloy formability carried out material characterization by means of the Hill-48 yield function and the Hollomon or Swift hardening laws, separately fitted for each sample direction [1,5,7,14,15,16]. Although the Hill-48 criterion is a simple anisotropic plastic model that requires a low number of material parameters, using independent models for the different loading directions drastically increases the number of material parameters. Additionally, the implementation of such an approach in some numerical codes could be cumbersome.

The use of the Hollomon hardening law simplifies the implementation of the constitutive model and the fitting process, showing good agreement between the experimental and numerical stress-strain curves, with less than 4 MPa of RMSE for the RD (see Table 5). It is important to point out that the change in mechanical response in directions other than RD is only driven by the yielding criterion CPB-06. In addition, as an interesting result, using the Hollomon hardening law identified from the RD data combined with the CPB-06 flow rule makes it possible to improve the fit of the hardening curve in the DD and TD. Thus, the RMSE in the DD and the TD is reduced by 50% in comparison to the one in the RD (see Table 5). This improvement can also be seen in a qualitative way when comparing the numerical predictions and experimental data in the stress-strain curve plotted in Figure 4. The good agreement shown in Figure 4 up to the Ultimate Tensile Stress (UTS) is an encouraging result, since it means that the damaging process of the material could eventually be captured by coupling the presented approach with some coupled non-local damage models [37,38,39]. 

An important feature of this approach is related to the change of the yielding locus induced by the CPB-06 yielding criterion. Figure 5 shows a comparison between the yielding surface in plane bi-axial stress (no shear) of different classic flow rules. The key features of the proposed approach are obviously the anisotropic nature of the yielding criterion and also the Strength Differential (SD) effect. In preliminary fitting runs, where the *k* parameter was set to 0 (neglecting SD effect), the error based on stress and Lankford increased in all three directions. especially for TD, where the fitted stress-strain curve was over-estimated for the entire range and the Lankford value decreased from 0.60 to 0.51 in TD. Assuming an asymmetric behavior (presence of SD effect) with a fitting of the *k* parameter, it is possible, at the same time, to match the stress-strain curves without compromising the estimation of the Lankford coefficients for all three directions (RD, DD and TD). 

It is also worth mentioning that the strain paths predicted by the model in the case of uniaxial loading present good agreement with the experimental data. Figure 6 shows the experimental and the numerical predictions of the strain path corresponding to the RD, the DD and the TD uniaxial tests. A more quantitative way of looking at the quality of the prediction in terms of transversal strain is to look at the different Lankford coefficients (*R*). It can be seen in Table 4 that the predicted *R* values are in adequate agreement with the corresponding experimental measurements given in Table 1. The relative error in each of the three Lankford coefficients is lower than 1%, which stems from the way they have been defined; thus, the RMSE is close to 0 in all three directions. The definition of the Lankford coefficients as functions of the *L*, *k* and *a* result in values that are almost the same as the ones determined experimentally. The previous results prove that the proposed approach is able to successfully predict the anisotropic mechanical response of the studied zinc alloy over different uniaxial loading directions. Furthermore, these classic and simple experimental tests provide all the information required in order to calibrate the model. However real-life applications involve mechanical loadings that are much more complex. For instance, biaxial loading conditions are common in many material forming industrial processes. The bulge test simulation (Section 3.2) is a very interesting application involving biaxial loading of the material sheet. The strain paths (experimental and numerical predictions) corresponding to the different elliptical dies used in the bulge test are plotted in Figure 6. On top of the aforementioned uniaxial strain paths, the numerical prediction of the equibiaxial loading condition also presents an excellent agreement with experimental measurements (see blue data series in Figure 6). Concerning the bulge tests with elliptical dies, they can be divided into two sets of experiments by using the material direction (RD or TD) oriented with the long axis of the ellipse. For the sake of simplicity, these two sets of bulge tests will be referred to as RD and TD, respectively.

The bulge RD tests present a slight deviation to the right of the experimental cloud point for the 0.5 die, but highly displaced to the left for the 0.33. In the case of the TD, the numerically obtained curves deviated slightly to the left for the 0.33 die but, contrary to the RD situation, were highly displaced for the 0.5 die. The slope of the different strain paths denotes the behavior described above. In particular, a satisfactory experimental validation of the numerical model was obtained for the tensile test, bulge equibiaxial and bulge for paths β = 0.50-RD and β = 0.33-TD, where only the cases β = 0.50-TD and β = 0.33-RD exhibit small differences.

## 5. Conclusions

The CPB-06/Hollomon associate constitutive model, in addition with the proposed fitting procedure, proves to be a valid and robust way to describe the elastoplastic anisotropic behavior of the Zn-20 alloy. In this context, a unique set of anisotropic coefficients was able to reproduce the experimental tensile stress-strain curves and Lankford coefficients. The strain paths in the bulge test using different dies were properly validated for the equibiaxial, β = 0.50-RD and β = 0.33-TD cases while only approximate results were obtained for the β = 0.50-TD and β = 0.33-RD cases. These results, together with the good approach of the stress-strain curves, reinforce the use of an associated flow rule to reproduce the anisotropy behavior of Zn-20 sheets. This improvement is closely related with the use of a specific yield function that considers the SD effect, as does the CPB-06. The use of an associated flow rule simplifies the implementation of the constitutive models, gives mathematical and physical consistency to the solution and reduces the complexity of the fitting process because a reduction in the number of coefficients to be defined. 

Finally, the present work sets new steps to improve the predictability of more general forming conditions including combined hardening laws and damage criteria.

## Figures and Tables

**Figure 1 materials-12-03072-f001:**
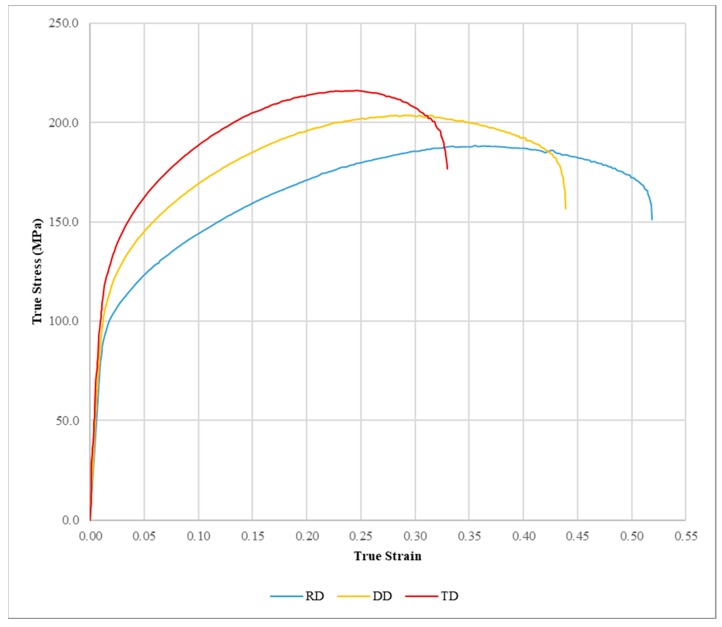
Experimental true stress-strain tensile curves for RD, DD and TD according to the data published in [1].

**Figure 2 materials-12-03072-f002:**
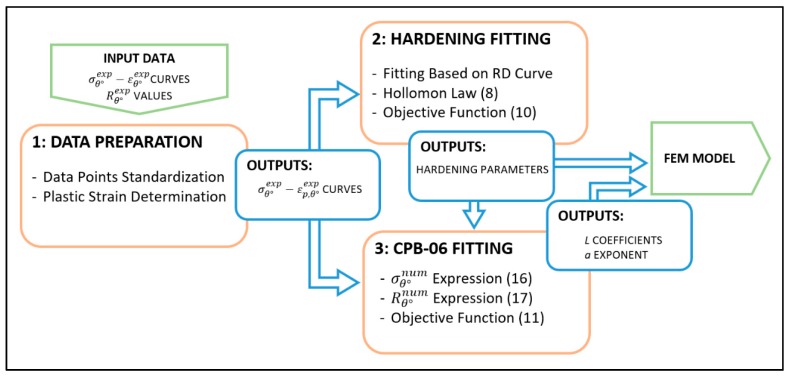
Flow diagram of the fitting procedure.

**Figure 3 materials-12-03072-f003:**
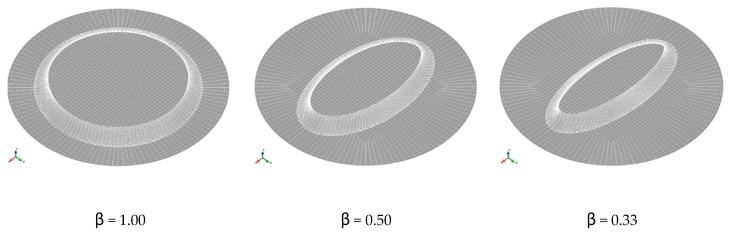
Geometrical models and finite element meshes of the bulge test for the different analyzed dies.

**Figure 4 materials-12-03072-f004:**
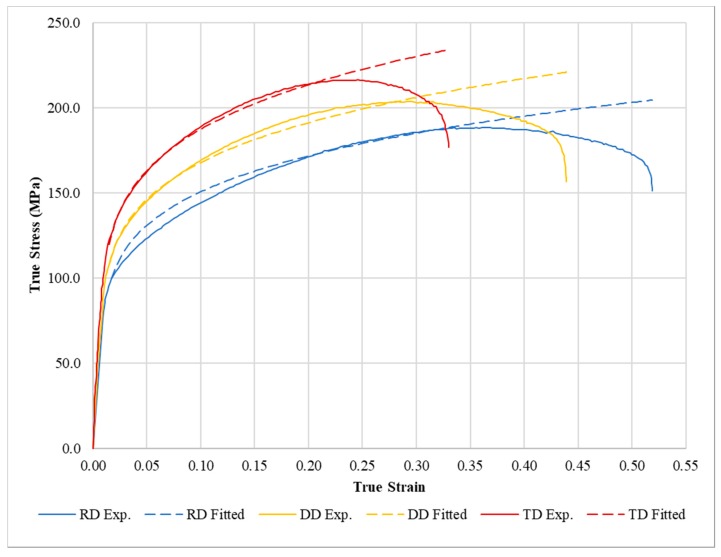
Experimental and adjusted true stress-strain tensile curves (for all cases, the fitted curves are plotted in the whole range of strain until fracture).

**Figure 5 materials-12-03072-f005:**
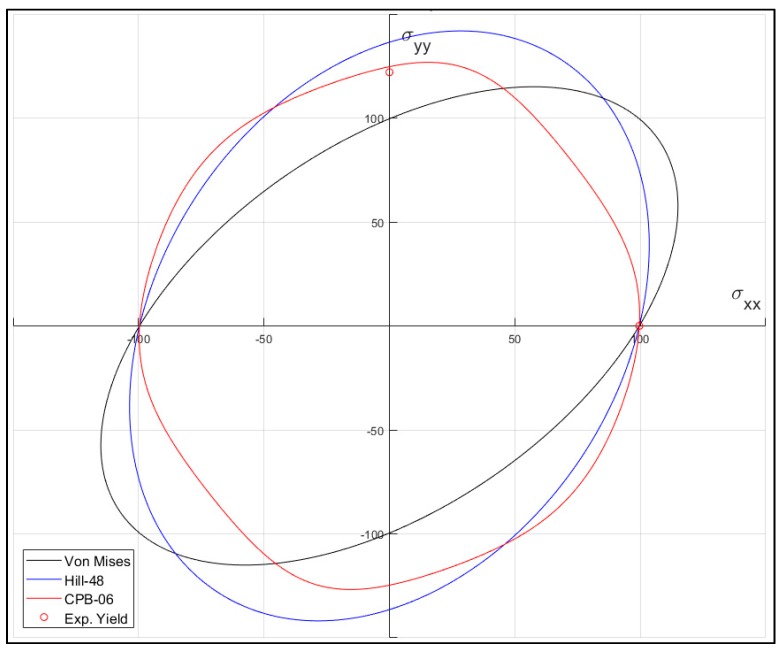
Plane stress yield envelope in the $\sigma_{xx}|\sigma_{yy}$ plane at the yield strength for the von Mises, Hill-48 and CPB-06 criteria (the red circles denote the yield strengths for each sample direction).

**Figure 6 materials-12-03072-f006:**
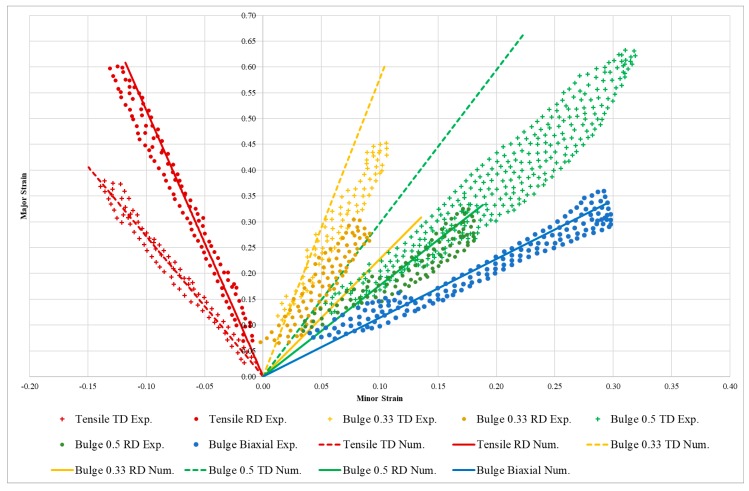
Experimental and numerical strain paths on the major and minor strains diagram from results of the tensile and bulge tests.

**Table 1 materials-12-03072-t001:** Experimentally measured mechanical properties of the Zn-20 alloy [1].

Parameter	RD	DD	TD
$\sigma_{yp}$ (MPa)	99.6	110.0	122.0
R	0.25	0.35	0.60

**Table 2 materials-12-03072-t002:** Hollomon fitted coefficients from the RD tensile curve.

*K* (MPa)	$\varepsilon_{0}$	n
171.38	0.363	0.538

**Table 3 materials-12-03072-t003:** CPB-06 fitted coefficients.

*L_12_*	*L_14_*	*L_22_*	*L_24_*	*L_33_*	*L_44_*	*a*
0.1011	−0.2115	0.9141	−0.0156	0.8408	1.0346	6.0

**Table 4 materials-12-03072-t004:** Numerical Lankford coefficients (R) and their relative errors.

	RD	DD	TD
R	0.25	0.35	0.60
Relative Error (%)	0.12	0.23	0.12

**Table 5 materials-12-03072-t005:** RMSE of the fitting procedure in the true stress-strain curves and Lankford coefficients.

	*RD*	DD	TD
$E_{\sigma }\;\left(\mathrm{MPa}\right)$	4.061	2.775	1.914
$E_{R}$	0.000	0.001	0.000

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
