# Peer review of "Characterization of the Elastoplastic Response of Low Zn-Cu-Ti Alloy Sheets Using the CPB-06 Criterion"

_materials, 2019, doi:10.3390/ma12193072_

Round 1
Reviewer 1 Report
see attached file

Author Response
Revised manuscript “Characterization of the elastoplastic response of low zinc alloy sheets using the CPB-06 criterion”, written by Francisco Alister et al., submitted to Materials (Manuscript ID: materials-578431). The authors gratefully acknowledge the reviewers´ valuable comments that have been incorporated into the revised version of the manuscript. The changes made in it are highlighted in red.
COMMENTS FOR THE AUTHOR:
Reviewer #1.
The authors present a calibration procedure for the CPB-06 yield surface and the Hollomon hardening law, which they apply to model Zn-Ti-Cu sheets.
Based on stress-strain data and r-values the authors claim that the chosen approach is especially suited for modelling Zn alloys. I do, however, see a number of issues in drawing this conclusion:
The fitting quality of the three stress-strain curves shown in Figure 4 is not that great after all. While the overall stress level is reasonably matched the hardening behaviour is not matched well at all. In this context I am very much surprised that even though the RD curve was used for parameter fitting it actually shows the worst agreement!
Authors' answer: As can be seen in Table 5, the root mean square error (RMSE) of the RD curve is effectively the worst. This is clearly a limitation of the Hollomon hardening law used in this work. Despite of this drawback, we still decided to employ this commonly used law (e.g., (Jansen et al., 2013; Milesi et al., 2017) because our final aim was to assess the capabilities of the CPB-06 yield criterion to deal with highly anisotropic materials like the alloy studied in this research.
Jansen, Y., R. Logé, M. Milesi, and E. Massoni. 2013. “An Anisotropic Stress Based Criterion to Predict the Formability and the Fracture Mechanism of Textured Zinc Sheets.” Journal of Materials Processing Technology 213(6): 851–55. http://dx.doi.org/10.1016/j.jmatprotec.2012.12.006. (reference [1] of the manuscript).
Milesi, M. et al. 2017. “Accounting for Material Parameters Scattering in Rolled Zinc Formability.” Journal of Materials Processing Technology 245: 134–48 (reference [5] of the manuscript).
Improving this law is definitely one of the important actions that will be carried out in the future.
To my knowledge the r-value or Lankford coefficient is defined as a strain ratio, however equation 17 sees to only use stress quantities? Could the authors please clearly state their definition of the r-value?
Authors' answer: We use the classical definition of the Lankford coefficients, i.e., the ratio between the strain increments along the width and thickness of the tensile sample. Considering the plastic flow rule (included at the end of Section 2.2) and the isochoric nature of the plastic flow (as stated in Section 2.3.3), the Lankford coefficients finally result in the expression shown in equation (17) (the same expression is reported in Cazacu and Barlat, 2004).
Cazacu, O., and F. Barlat. 2004. “A Criterion for Description of Anisotropy and Yield Differential Effects in Pressure-Insensitive Metals.” International Journal of Plasticity 20(11 SPEC. ISS.): 2027–45.
The authors state that their description is much better suited than others. However, they fail to actually show that. All they show is a comparison of yield surface sections in Figure 5. While these are clearly different this does not necessarily imply they are worth. Moreover the authors choose von Mises and Hill48 as comparison. For both of them I would not expect a good description of hcp materials, as von Mises is isotropic and Hill48 shows a fourfold symmetry!
Authors' answer: It is the authors´ intention to estimate the hardening behaviour of the alloy with the use of an associated plasticity model. The comparison of the yield surface was included to make apparent the potential predictive capability of the CPB-06 criterion in comparison to the well-known Hill-48 criterion using, of course, the same experimental tensile curve for both (the von Mises yield function was only added as a reference). It is seen that the Hill-48 criterion poorly represents the anisotropy of the studied zinc alloy. Moreover, and based on the literature review, an associated Hill-48 criterion is not in general able to simultaneously fit the stress-strain curves and the Lankford coefficients with a unique set of hardening and anisotropic parameters.
One key argument of the authors why CPB-06 is especially well suited is the fact that it accounts for the Strength Differential Effect. However, on page 6 the authors clearly state that they set k=0! Also the yield surface plotted in figure 6 does not show any asymmetry.
Authors' answer: Because the work used an existing data base, we don´t have experimental data to assume in this case an asymmetric behaviour of the alloy. Due to this, the symmetric form of the CPB-06 criterion was used, i.e., k=0 (the assessment of the Strength Differential Effect in this alloy definitely constitutes a challenging task and is one of the aims of a research that we are currently carrying out). As already mentioned, the main interesting feature of using the CPB-06 criterion in this work relies on its ability to properly estimate the plastic behaviour of the studied alloy with one set of hardening and anisotropic parameters.
Finally the authors themselves state that the prediction of bulge test data is deviating quite a bit from the experimental data with no visible systematic.
Authors' answer: Specifically, a satisfactory experimental validation of the numerical model was obtained for tensile test (RD and TD), bulge equibiaxial and bulge for paths β=0.50-RD, where only the cases β=0.50-TD and β=0.33-TD exhibit small differences.
Technical issues:
Are the stress-strain curves shown in figure 1 really true stress vs. true strain? To me they look rather like technical stress-strain curves.
Authors' answer: The curves published in the referred paper [1] are true stress-equivalent strain (e1) according to the expressions described therein. We transform them to true stress-true strain curves through the conversions also reported in [1].
Some of the symbols in figure 6 are hard to see and the legend is too small to read.
Authors' answer: Improved.
In summary, what remains from this manuscript is the fitting of a known yield surface description using data of a not so commonly used material. The overall fitting quality is ok but to claim the scheme as especially suited a rigid comparison with other possibly suited advanced yield surface descriptions, e.g. accounting for kinematic hardening, would be needed.
The manuscript in its current form would need a major revision to be accepted for publication in materials.
Additional comments:
English language and style are fine/minor spell check required. Done.
Are the methods adequately described? Improved.
Are the results clearly presented? Improved.
Are the conclusions supported by the results? Improved.

Reviewer 2 Report
This manuscript has been well prepared. Authors have constructed The CPB-06/Holloman associate constitutive model. They have proved the validation of fitting process for describing the elastoplastic anisotropic behaviour of the Zn-20 alloy.
They have managed with this work to reproduce the experimental tensile stress-strain curves and Lankford coefficients. The strain paths in the bulge test using different dies were generally validated for the different directions.
The authors have improved the predictability of more general forming conditions including combined hardening laws and damage criteria.
I suggest to be published in the actual form in this journal.
Author Response
Revised manuscript “Characterization of the elastoplastic response of low zinc alloy sheets using the CPB-06 criterion”, written by Francisco Alister et al., submitted to Materials (Manuscript ID: materials-578431). The authors gratefully acknowledge the reviewers´ valuable comments that have been incorporated into the revised version of the manuscript. The changes made in it are highlighted in red.
COMMENTS FOR THE AUTHOR:
Reviewer #2.
This manuscript has been well prepared. Authors have constructed The CPB-06/Holloman associate constitutive model. They have proved the validation of fitting process for describing the elastoplastic anisotropic behaviour of the Zn-20 alloy.
They have managed with this work to reproduce the experimental tensile stress-strain curves and Lankford coefficients. The strain paths in the bulge test using different dies were generally validated for the different directions.
The authors have improved the predictability of more general forming conditions including combined hardening laws and damage criteria.
I suggest to be published in the actual form in this journal.

Reviewer 3 Report
In this study, the associated form of the CPB-06 yield criterion together with the Hollomon hardening law has been implemented in a FEM code for the estimation of the elastoplastic response of a low Zn-Ti-Cu alloy. The results have been confirmed with the data extracted from a published work. The model is interesting and can be used for the prediction of the yield, however, the manuscript has some problems that must be addressed before acceptance for the publication in Materials.
Authors have used CPB-06 yield function term in the abstract without any description before that. Sentences 101-105 should be rewritten in a way that becomes clear to the readers that tensile tests has not been performed by the authors and just extracted from a publication. In reference [1], the chemical composition of the alloy could not be found. In this case, how the authors did use the tensile properties of that alloy in this research? There is the same issue for the elastic properties extracted from [30] In my opinion, using just one alloy and thickness, to confirm the model is not accurate and need to add more experimental data to confirm the model, or at least change the title of the manuscript by replacing the general term of low zinc alloy sheets to a specific zinc alloy. Although the importance and the novelty of this work may be clear for a researcher in the field of mechanical engineering, the significance and originality of this research is not that much clear for a material scientist (who are the most of the readers of “Materials”). Therefore, it is essential to add a couple of sentences to the introduction as well as the first paragraph of section 2.2 to clarify these to the readers. It is not clear that which of the equations are the original one and which are referenced from the previous works. Please check all the equations to be sure that those need to be referenced are referenced. Hollomon fitted coefficients in Table 5 are calculated for TD, DD or RD? If it is possible please add the finite element codes in a supplementary file. Which software or Program did you use for the FEM analysis?
Author Response
Revised manuscript “Characterization of the elastoplastic response of low zinc alloy sheets using the CPB-06 criterion”, written by Francisco Alister et al., submitted to Materials (Manuscript ID: materials-578431). The authors gratefully acknowledge the reviewers´ valuable comments that have been incorporated into the revised version of the manuscript. The changes made in it are highlighted in red.
COMMENTS FOR THE AUTHOR:
Reviewer #3.
In this study, the associated form of the CPB-06 yield criterion together with the Hollomon hardening law has been implemented in a FEM code for the estimation of the elastoplastic response of a low Zn-Ti-Cu alloy. The results have been confirmed with the data extracted from a published work. The model is interesting and can be used for the prediction of the yield, however, the manuscript has some problems that must be addressed before acceptance for the publication in Materials.
Authors have used CPB-06 yield function term in the abstract without any description before that. Sentences 101-105 should be rewritten in a way that becomes clear to the readers that tensile tests has not been performed by the authors and just extracted from a publication. In reference [1], the chemical composition of the alloy could not be found. In this case, how the authors did use the tensile properties of that alloy in this research? There is the same issue for the elastic properties extracted from [30] In my opinion, using just one alloy and thickness, to confirm the model is not accurate and need to add more experimental data to confirm the model, or at least change the title of the manuscript by replacing the general term of low zinc alloy sheets to a specific zinc alloy. Although the importance and the novelty of this work may be clear for a researcher in the field of mechanical engineering, the significance and originality of this research is not that much clear for a material scientist (who are the most of the readers of “Materials”). Therefore, it is essential to add a couple of sentences to the introduction as well as the first paragraph of section 2.2 to clarify these to the readers. It is not clear that which of the equations are the original one and which are referenced from the previous works. Please check all the equations to be sure that those need to be referenced are referenced. Hollomon fitted coefficients in Table 5 are calculated for TD, DD or RD? If it is possible please add the finite element codes in a supplementary file. Which software or Program did you use for the FEM analysis?
Authors' answer: The reviewer’s concerns are pertinent; we appreciate the suggestions. The reviewer´s remarks are commented separately below.
The name of the authors of the CPB-06 yield function was included in the abstract.
Sentences 101-105 have been slightly rewritten to clearly state that the experimental curves presented in Figure 1 are those obtained and reported in [1].
The chemical composition was obtained from the doctoral thesis of the first author of [1]. Since this information does not contribute to the global aim of the paper, it was removed in the revised version of the manuscript.
As described in Section 2.3, we use the experimental values of the axial true stress-strain curves (until the Ultimate Tensile Stress to guarantee homogeneous stress and strain fields) to calibrate the model. Although the characterization is mainly focused on the plastic response of the alloy, note that its elastic behaviour is also considered through equation (9), assuming an isotropic elastic response.
In the revised version of the manuscript, the title of the manuscript was change to emphasize that we are exclusively dealing with a specific low Zn-Ti-Cu alloy.
The equations included in the manuscript have been already reported elsewhere. The main original contribution of this work is the application of the CPB-06 model to the characterization of the Zn-20 alloy. In addition, one important difference between our work with the referenced studies of Jansen, Milesi and coworkers is that the proposed approach allows the fitting of all directions without the definition of a particular set of parameters angle dependent.
The numbering of the equations were revised and corrected.
The Hollomon fitted coefficients in Table 5 were calculated from the DD tensile curve. This was said in Section 2.3.2 and, in the revised version of the manuscript, it was also added in the caption of Table 5.
As mentioned at the end of Section 2.2, we used for the simulations presented in this work an in-house FE code which was extensively used and validated in many engineering applications. We do not add all the references in which this code was used so as not to divert the attention from the focus of this work but, of course, a list of publications reporting numerical results obtained with such code can be added as supplementary material.
Additional comments:
English language and style are fine/minor spell check required. Done.
Does the introduction provide sufficient background and include all relevant references? Improved.
Is the research design appropriate? Improved.
Are the methods adequately described? Improved.
Are the results clearly presented? Improved.
Are the conclusions supported by the results? Improved.

Round 2
Reviewer 1 Report
see attached review file

Reviewer 3 Report
The manuscript has been revised moderately. Although all of the reviewers' comments have not addressed, the manuscript has been improved and can be published in the present format.